The effect of the Wnt pathway on the osteogenic differentiation of periodontal ligament stem cells in different environments

http://orcid.org/0009-0003-7864-8819 Su Qi 1
Huang Fengqiong 2
Fang XiuLei 2
http://orcid.org/0009-0002-8602-1293 Lin Qiang 2 qianglin1991@jnu.edu.cn
1 College of Stomatology, Jinan University , Guangzhou, Guangdong , China
2 Hospital of Stomatology, The First Affiliated Hospital of Jinan University , Guangzhou, Guangdong , China
Gould Gwyn
Electronic publication date: 2025 Jan 3
Publication date: 2025
Volume: 13
Electronic Location ID: e18770
Received 2024 Aug 12; Accepted 2024 Dec 5
Copyright: © 2025 Su et al.
Copyright year: 2025
Copyright holder: Su et al.
License: This is an open access article distributed under the terms of the Creative Commons Attribution License, which permits unrestricted use, distribution, reproduction and adaptation in any medium and for any purpose provided that it is properly attributed. For attribution, the original author(s), title, publication source (PeerJ) and either DOI or URL of the article must be cited.
License URL: https://creativecommons.org/licenses/by/4.0/

Keywords: Periodontal ligament stem cells, Osteogenic, Wnt

Funding: Guangdong Medical Research Fund Project A2023150 Guangzhou Science and Technology Plan Project, China 2024A03J0821 This research was funded by the Guangdong Medical Research Fund Project (A2023150) and the Guangzhou Science and Technology Plan Project, China (2024A03J0821). The funders had no role in study design, data collection and analysis, decision to publish, or preparation of the manuscript.

==============================
Alveolar bone defects have always been an urgent problem in the oral cavity. For some patients with periodontal disease or undergoing orthodontic treatment or implant restoration, alveolar bone defects can greatly inconvenience clinical diagnosis and treatment. Periodontal ligament stem cells (PDLSCs) are considered a promising source for stem cell therapy due to their high osteogenic differentiation capability. The osteogenic differentiation of PDLSCs is influenced by various cytokines mediating signaling pathways, as well as a range of biochemical, physical, and chemical factors. Different environments typically impact the Wnt signaling pathway and consequently affect the osteogenic differentiation of PDLSCs. Understanding the effects of the Wnt signaling pathway on the osteogenic differentiation of periodontal ligament stem cells in various environments contributes to addressing the diverse clinical scenarios encountered in the oral environment. This knowledge aids in influencing the pathway through bioengineering or the development of novel materials, thereby enhancing the osteogenic potential of stem cells and providing new diagnostic and therapeutic strategies for clinical practice.

Introduction

Alveolar bone defects have always been a common problem in the oral cavity. Maxillofacial trauma and alveolar bone resorption cause bone defects, which make maxillofacial surgery and periodontal treatment difficult. Stem cell therapy has been suggested as a research focus in tissue engineering. However, stem cells are limited in their research and application, and so having a large number of stem cells is very important when prioritizing efficiency. Periodontal ligament stem cells (PDLSCs) are a kind of undifferentiated mesenchymal cells that are mainly extracted from the periodontal ligament and have the potential to differentiate into osteoblasts, adipocytes, chondrocytes, and other cell types in vitro (Xue, Yu & Chen, 2018). They have the potential for a broad application in periodontal tissue regeneration repair, orthodontic treatment, and implant/biomaterial repair. The osteogenic differentiation potential of PDLSCs has attracted much attention in these applications. The osteoblastic differentiation potential of PDLSCs has attracted extensive attention in these applications. The osteoblastic differentiation process is influenced by various signaling pathways, among which the Wnt signal and its regulatory factors play an important role in tissue repair, regeneration, and regulation of growth and differentiation. Wnt signals also regulate the proliferation and differentiation of PDLSCs, and their effects are related to the environment and differentiation stage (Bao et al., 2021).

In 2004, the first batch of PDLSCs were isolated from periodontal tissues by Seo et al. (2004) using cloning technology. These unique stem cells are abundant in periodontal tissues and display a remarkable potential for osteogenic differentiation (Lindroos et al., 2008). Studies have demonstrated that PDLSCs exhibit an increased formation of calcified nodules compared to gingival and dental pulp stem cells, indicating their superior osteogenic differentiation ability and suitability for studies focusing on bone regeneration (Lindroos et al., 2008).

Moreover, PDLSCs have been shown to possess a higher growth potential and enhanced proliferation ability when compared with human bone marrow-derived MSCs and dental pulp stem cells (Lei et al., 2021). Their ease of isolation from periodontal tissues during routine dental cleaning and root planing procedures, as well as their collection from discarded biological samples in dental clinics, offer a non-traumatic and convenient method for harvesting these valuable stem cells (Tassi et al., 2017; Iwata et al., 2010). Consequently, PDLSCs have emerged as a prominent candidate for applications in periodontal tissue regeneration, repair, and tissue engineering.

Furthermore, studies involving the in vitro culture of PDLSCs with osteogenic induction solutions have revealed a significant increase in the expression of specific markers such as Runx2, bone morphogenetic protein 2 (BMP2), alkaline phosphatase (ALP), and BMP2 (Mo et al., 2022). These markers are widely recognized as crucial indicators of the osteogenic differentiation potential of PDLSCs. Additionally, recent research has illuminated the complex regulatory network governing the differentiation of PDLSCs into osteoblasts, implicating various environmental factors and signaling pathways such as the Wnt, TGF-β/BMP, and P38/JNK pathways. Targeted genes or proteins such as microRNAs (miRNAs/miRs), long non-coding RNAs (lncRNAs), and circular RNAs (circRNAs) have also been identified as key modulators in regulating the expression of osteogenic genes in PDLSCs (Wang et al., 2022b) (Fig. 1).

Figure 1 PDLSC source and multidirectional differentiation ability.

PDLSCs enhance osteogenic differentiation through ncRNA and Wnt signaling pathways and are expected to be applied in oral clinical practice. Created in BioRender.

Depending on the role of β-catenin, Wnt signaling pathways can be divided into classical Wnt/β-catenin channels and non-classical Wnt/Ca2+ channels (Rim, Clevers & Nusse, 2022). When various secreted proteins such as Wnt1, Wnt3a, and Wnt5a bind to the Frizzled receptor, the Dishevelled protein is activated, which inhibits the active glycogen synthase kinase-3β (GSK-3β), allowing beta-catenin to accumulate in the cytoplasm. Subsequently, β-catenin is transported into the nucleus where it binds to the T cell-specific factor/lymphocyte enhancer factor (TCF/LEF) and lymphocyte enhancer factor (LEF) to form a complex, thus creating a typical Wnt/β-catenin pathway. This pathway is crucial for regulating the transcription of Wnt target genes and participating in bone growth (Wang et al., 2022a).

On the other hand, the non-classical Wnt signaling pathway activates the heterotrimeric G protein by binding the Wnt protein to its receptor, leading to the activation of phospholipase-C (PLC) and resulting in increased intracellular Ca2+ release. The activation of Ca2+-calmodulin-dependent protein kinase-II (CaMKII) or protein kinase-C (PKC) and calcineurin (Caln) simultaneously inhibits the transcription of β-catenin by activated T nuclear factor (NFAT), thereby antagonizing the classical Wnt signaling pathway (Maeda et al., 2019) (Fig. 2).

Figure 2 Classical and non-classical Wnt signaling pathways.

Wnt1, Wnt3a, and Wnt5a bind to the receptor Frizzled to activate the Dishevelled protein and inhibit the active GSK-3β, allowing beta-catenin to accumulate in the cytoplasm. Subsequently, β-catenin is transported into the nucleus and bound to TCF/LEF and LEF to form a complex, producing a typical Wnt/β-catenin pathway, which is used to regulate the transcription of Wnt target genes and participate in bone growth. Created in BioRender.

The main focus of this article is the challenging issue of alveolar bone defects in patients with periodontal disease who are undergoing orthodontic treatment or seeking implant restoration in clinical practice. When employing stem cell therapy, it is essential to fully consider the signaling pathways that influence the osteogenic differentiation of stem cells in various environments. This allows for targeted therapy aimed at promoting alveolar bone regeneration. Current research attempts to influence the osteogenic differentiation of periodontal ligament stem cells through micro-signaling pathways or to explore the mechanisms of these pathways in different environments. However, there is a lack of suitable carriers or products for clinical application. This article aims to integrate existing studies on the effects of the Wnt signaling pathway on the osteogenic differentiation of periodontal ligament stem cells in various environments. It seeks to bridge the information and knowledge gaps across disciplines, providing a theoretical basis for future applications that micro-engineer macro-level outcomes. The goal is to develop novel biomaterials or targeted drugs that precisely act on the pathways promoting stem cell osteogenesis, thereby addressing the challenges of bone defect repair in clinical dental practice, which often involve significant difficulty and prolonged treatment times.

The common oral environment during the clinical diagnosis and treatment process of the oral cavity acts as the linking factor between different environmental conditions; for instance, patients with alveolar bone defects caused by periodontitis, those who have experienced secondary alveolar bone loss after implant surgery, and patients with bone defects such as bone window cracking during orthodontic treatment. Patients experience diverse oral environments that influence the osteogenic differentiation of PDLSCs and consequently impact the effectiveness of stem cell therapy.

Understanding the relationship between the environment and signaling pathways enables the development of tailored measures for each pathway target. This involves the preparation of specific drugs to enhance the osteogenic differentiation of stem cells, leading to controllable and targeted alveolar bone regeneration. This approach aims to alleviate the suffering of patients.

Currently, a significant issue faced by both clinicians and patients is the difficulty, prolonged duration, painful process, and variability in morphology associated with the repair of bone defects. These clinical challenges greatly hinder the advancement of bone defect repair. Although bone grafts are widely used in oral clinical practice, they still exhibit the aforementioned drawbacks. Additionally, clinicians encounter difficulties in providing more specific and effective products that can non-invasively enhance the bone remodeling process for patients with periodontal disease or those undergoing orthodontic treatment.

Therefore, there is a strong expectation for the development and application of non-invasive bioengineering therapies or novel implant materials to address these clinical challenges while improving the patient experience during diagnosis and treatment. Achieving this goal requires interdisciplinary collaboration between clinicians and researchers to integrate micro-level biological pathways with macro-level targeted products, thereby efficiently promoting the osteogenic differentiation of periodontal ligament stem cells and facilitating targeted alveolar bone regeneration.

Investigation methods

The following four databases were thoroughly searched to find all pertinent studies: MEDLINE (via PubMed), Embase, the Cochrane Library, and Web of Science. In addition, a manual search was done by looking through the associated articles’ reference lists. Articles, reviews, editorials, research letters, and systematic reviews that covered all of the subjects included in the review, such as the Wnt signaling pathway and osteogenic differentiation of periodontal ligament stem cells, met the inclusion criteria for this review. “Wnt signaling pathway” and “Osteogenic differentiation of periodontal ligament stem cells” were the keywords used in the search approach. The search strategy was as follows: “(“Wnt signaling pathway” [Mesh]) and ((Osteogenic differentiation of periodontal ligament stem cells) or (Osteogenic differentiation))” and “(“Wnt signaling pathway” [Mesh]) and (“Osteogenic differentiation” [Mesh]),” which was developed for MEDLINE and modified for the other databases. A systematic literature search is conducted by incorporating various keywords such as “inflammatory microenvironment,” “orthodontics,” and “implantation.” To ensure the relevance and quality of the articles, several filters are applied during the search process. Articles published before 2010 are excluded to focus on the most recent advancements in the field. Additionally, articles published in languages other than English are filtered out to facilitate a more straightforward review process.

Priority is given to articles published in high-impact journals within the fields of cell biology, regenerative medicine, and dentistry. Journals such as “Stem Cells,” “Journal of Bone and Mineral Research,” and “Tissue Engineering” are specifically included in the search criteria, while lesser-known or predatory journals are excluded to maintain the integrity of the research.

In summary, a systematic approach is employed to search for literature on the Wnt signaling pathway and the osteogenic differentiation of periodontal ligament stem cells, utilizing targeted keywords and strategic filtering based on publication year, language, and journal quality. This method ensures the collection of a robust set of relevant and high-quality research articles for further analysis.

Wnt signaling pathway is involved in the osteogenic differentiation of pdlscs

Wnt signaling plays a crucial role in the early development, organ formation, tissue regeneration, and various physiological processes in animal embryos. It regulates stem cell proliferation, differentiation, migration, polarity, and renewal by activating diverse intracellular signaling cascades. The Wnt pathway is involved in the regulation of the cell cycle, cell growth, and differentiation, and is essential for maintaining bone homeostasis, which is critical for periodontal regeneration (Wei et al., 2021). When the Wnt signaling pathway is activated, osteoclast generation is inhibited, alveolar process formation is enhanced, and the width of the periodontal membrane is reduced (Yang et al., 2020; Lim et al., 2015). Thus, the Wnt pathway is closely associated with bone formation.

Multiple studies have demonstrated that the Wnt/β-catenin pathway is involved in the osteogenic differentiation of PDLSCs (Bao et al., 2021; Wang et al., 2021). Researchers stimulated human PDLSCs (hPDLSCs) with 10 mM lithium chloride (LiCl), a typical activator of the Wnt signaling pathway, and observed a positive effect on the osteogenic differentiation of these cells (Liu et al., 2019). Additionally, they inhibited the Wnt/β-catenin signaling pathway using a β-catenin blocker, which resulted in a 25% reduction in the OPG/RANKL ratio. The gene expression levels of osteogenic markers, including ALP, BMP2, and Runx2, were also down-regulated, suggesting the involvement of the Wnt signaling pathway.

Further studies identified recombinant human Notum, a phospholipase shed on the cell surface, as a novel negative regulator of osteogenic differentiation in hPDLSCs under osteogenic induction conditions (Yang et al., 2021). The application of LiCl following Notum treatment revealed that LiCl significantly mitigated Notum’s inhibitory effect on the osteogenic differentiation of hPDLSCs. Combining these findings with previous studies suggests that the Wnt signaling pathway plays a crucial role in the osteogenic differentiation of hPDLSCs.

Wnt pathway and aging of pdlscs cells

Activation of the Wnt/β-catenin signaling pathway serves as a common mechanism that regulates cell differentiation, preventing bone aging and inflammation (Yan et al., 2015; Yu et al., 2014). The decline in stem cell viability and the aging process of stem cells can both impact the osteogenic differentiation potential of these cells (Zhu et al., 2023). Recent research has established a close connection between the Wnt signaling pathway and stem cell aging, indicating that aging might disrupt the regulatory role of the classical Wnt/β-catenin signaling pathway in the osteogenic differentiation of PDLSCs (Du, 2018).

Studies have shown that the natural extracellular matrix (ECM) can rejuvenate aging cells, preserving their differentiation potential through the Wnt pathway (Zhang et al., 2018). Hence, the Wnt signaling pathway not only influences cell viability but also holds significant importance in the rejuvenation of senescent cells and maintenance of the osteogenic differentiation capacity of stem cells.

Different environmental regulation of the wnt signaling pathway affects pdlscs osteogenic differentiation

In the realm of clinical diagnosis and treatment, addressing alveolar bone defects stands as a pressing challenge in oral healthcare. The osteogenic differentiation potential of PDLSCs offers a novel therapeutic approach to tackle this issue. Nevertheless, various oral environments exert influence on the differentiation capacity of stem cells. Exploring how factors such as inflammation, orthodontic stress, and biomaterials impact the osteogenic differentiation potential of PDLSCs via the Wnt pathway is essential. This understanding can aid in optimizing the osteogenic capabilities of stem cells based on pathway responses to diverse conditions and facilitating the development of innovative clinical treatment strategies.

Inflammatory microenvironment

Periodontitis is a prevalent chronic infectious disease characterized by the action of inflammatory factors that can lead to clinical manifestations such as alveolar bone resorption and tooth mobility. Bacterial plaque associated with periodontitis commonly produces lipopolysaccharide (LPS), which subsequently stimulates host cells to generate inflammatory factors (Xie et al., 2021). Notably, tumor necrosis factor α (TNF-α) and interleukins (ILs) are expressed at elevated levels. When these inflammatory factors become excessive or dysregulated, various inflammatory diseases, including periodontitis, may arise.

Tissue chemistry analyses indicate that the expression of markers related to the osteogenic differentiation of PDLSCs diminishes within an inflammatory microenvironment (Lin et al., 2023). Under inflammatory conditions, PDLSCs exhibit enhanced proliferation and shorter doubling time compared to other stem cells (Fawzy El-Sayed et al., 2019). PDLSCs interact with the surrounding inflammatory stimuli through their surface receptors, demonstrating a robust resistance to inflammatory stress (Calabrese, 2022). Numerous studies have utilized exosomes extracted from various sources to stimulate PDLSCs in efforts to repair defects within the periodontitis context (Hu et al., 2023; Qiao et al., 2023).

However, in the local periodontitis microenvironment, inflammatory factors, especially TNF-α, tend to inhibit the osteogenic differentiation of PDLSCs (Lin et al., 2023). This inhibition results in a reduced expression of osteogenic differentiation-related markers in PDLSCs, thereby limiting the application of PDLSCs for the repair of periodontal bone defects to a certain extent. Consequently, understanding how the inflammatory microenvironment impacts osteogenesis through the common osteogenic signaling pathway, Wnt, is essential. This understanding will provide a more reliable theoretical foundation for future research directions and improvement strategies.

In the inflammatory microenvironment, both classical and non-classical Wnt signaling pathways maintain a relatively stable state. The classical Wnt pathway can have either osteoinductive or inhibitory effects, depending on the degree of signal transduction. It primarily operates through the non-canonical Wnt signaling pathway to stimulate CaMKII activation. This activation prevents β-catenin-related physiological activities, thereby inhibiting the transcription of Wnt target genes and adversely affecting osteogenesis (Qian et al., 2021). Furthermore, various inflammatory factors produced during the inflammatory response can also disrupt the Wnt/β-catenin signaling pathway, further influencing the potential for osteogenic differentiation.

TNF-α affects bone differentiation of PDLSCs through the Wnt pathway

TNF-α, a recognized pro-inflammatory cytokine involved in the inflammatory immune response, plays a crucial role in regulating various cell functions in the body. A clinical study demonstrated that the expression of TNF-α is elevated in the gingival crevicular fluid of patients with chronic periodontitis (Duarte et al., 2019). Additionally, the osteogenic differentiation ability of PDLSCs in chronic periodontitis patients was significantly lower compared to healthy PDLSCs (Wu et al., 2024). Scholars suspect that TNF-α may play a significant role in interfering with the osteogenic differentiation of PDLSCs. Researchers found through experimental comparisons that TNF-α is a key inflammatory factor responsible for reducing the osteogenic ability of periodontal membrane stem cells (Jin et al., 2022). Furthermore, in the presence of inflammatory conditions, the phosphorylation of GSK-3β in PDLSCs was notably increased, along with elevated expression of β-catenin, suggesting a reduction in osteogenic differentiation potential mediated by the Wnt/β-catenin signaling pathway (Cheng & Zhou, 2020). The studies referenced involved the transfection of PDLSCs in a high-dose TNF-α environment where β-catenin content was down-regulated using siRNA (Kong et al., 2015). These studies reported that the osteogenic ability of the stem cells was restored, and the negative regulatory effect of TNF-α on the osteogenic differentiation of PDLSCs was reversed. This finding suggests that manipulating β-catenin levels could potentially counteract the inhibitory effects of TNF-α on osteogenic differentiation in PDLSCs (Liu et al., 2014).

It’s suggested that TNF-α exerts its effects by influencing β-catenin to activate the classical Wnt signaling pathway. Previous studies have shown that TNF-α can modulate GSK-3β and β-catenin to regulate the osteogenic differentiation potential of PDLSCs. Therefore, the regulation of the Wnt signaling pathway, particularly the Wnt/β-catenin pathway, by TNF-α can significantly impact the osteogenic potential of human periodontal stem cells. This underscores the importance of controlling the microenvironment in periodontal tissue engineering and offers a new direction for treating periodontal inflammation by potentially inhibiting the TNF-α signaling pathway in the future.

Interleukin affects osteogenic differentiation of PDLSCs through Wnt pathway

The interleukin family consists of multiple subtypes of cytokines that play a crucial role in immune response and bone metabolism (Yang et al., 2021). Inflammatory responses can lead to the production of inflammatory cytokines such as IL-1, IL-6, and IL-17. Studies have shown that IL-1β affects PDLSCs in a dose-dependent manner, with low doses (0.01 ng/ml) promoting osteogenic differentiation while high doses (>0.1 ng/ml) inhibit by affecting the Wnt signaling pathway (Zhou et al., 2020; Hu et al., 2019).

Similarly, IL-17 can have both anti-osteogenic and osteogenic effects, depending on the cell source and culture conditions. Inflammatory environments can decrease the osteogenic ability of PDLSCs, partly due to the promotion of the IKK-NF-κB complex by a large amount of IL-17, resulting in the degradation of β-catenin and inhibition of the downstream process mediated by Wnt signaling molecules (Krstić et al., 2021; Chen et al., 2013). In addition, the phenomenon that IL-33 in the interleukin family significantly stimulated the expression of NF-κB and inhibited the expression of β-catenin at all test time points also indicated that it regulated the osteogenic differentiation of PDLSCs by affecting the Wnt /β-catenin pathway (Kukolj et al., 2019). Therefore, in the inflammatory microenvironment, interleukin can further change the osteogenic differentiation potential of PDLSCs through the Wnt signaling pathway, and reasonable control of the range of interleukin in the periodontitis environment will help to maximize the osteogenic ability of PDLSCs.

Orthodontic stress

PDLSCs have irreplaceable advantages in the orthodontic environment. When teeth are subjected to mechanical forces like chewing, occlusal contact, and orthodontic tooth movement, it is PDLSCs within the periodontal ligament that are the initial responders to such forces. A large number of studies have shown that orthodontic stress loaded with mechanical stress can accelerate the osteogenic differentiation of PDLSCs (Pakpahan et al., 2024; Zhang et al., 2023a; Shao et al., 2023). However, it is worth noting that most of the studies are limited to the laboratory environment, which leads to some different conclusions due to the different time and method of modeling. There is a lack of clinical trials on the effect of orthodontic stress on PDLSC osteogenesis, making it worthy of further exploration in the future.

During orthodontic treatment, teeth undergo a series of changes due to corrective force. External mechanical stress stimulation can activate the Wnt/β-catenin signaling pathway in PDLSCs through Akt phosphorylation (He et al., 2016), which converts physical signals into chemical signals and thus produces biological effects. Orthodontic stress can induce nuclear translocation of intracellular β-catenin, thereby stimulating the expression of nuclear transcription factor Tcf, activating Wnt/β-catenin signaling pathway (Yoo et al., 2018), promoting the transcription of downstream target genes, and thus promoting osteogenic differentiation of PDLSCs. In vitro studies have shown that the Wnt/β-catenin pathway is activated by compressive loading in PDLSCs (Gortazar et al., 2013), an in vivo study showed that β-catenin expression increased first and then decreased on the pressure side during orthodontic tooth movement in rats, and Wnt3a and Wnt10b levels increased on the compression side starting from day 5 (Gortazar et al., 2013). Some scholars (Zhang et al., 2016) also used Wnt/β-catenin pathway inhibitors in orthodontic animal models, and found that ALP decreased by 10 U/g, and Runx2 protein and mRNA expression levels decreased. The inhibited Wnt/β-catenin pathway in PDLSCs had altered stress effects on osteogenic differentiation and RANKL/OPG ratio. This further confirmed that the classical Wnt pathway mediates hydraulic signal transduction in PDLSCs and regulates osteogenic differentiation through its downstream genes, which may play an important role in periodontal tissue remodeling during orthodontic treatment.

Studies have found that orthodontic stress can promote the early osteogenic differentiation of hPDLSCs (Li, 2019). In the orthodontic stress area, the hypoxic environment has been found to enhance the osteogenic differentiation capability of stem cells. This hypoxic condition induces the activation of the Wnt/β-catenin signaling pathway in these stem cells (Xiao, 2020). These results suggest that hypoxia has a correlation effect on the expression of Wnt signaling pathway. Some scholars have found that osteogenic markers of stem cells are all upregulated under hypoxia, and the less oxygen, the greater the degree of upregulation. Hypoxia can promote osteogenic differentiation and induce activation of Wnt/β-catenin signaling pathway (Xiao et al., 2017), which is also similar to the findings of Li et al. (2023). The mRNA and protein levels of AXIN2, β-catenin, c-myc, and other genes associated with the Wnt signaling pathway are assessed using qPCR and Western blot, and the results reveal a significant upregulation of these genes’ mRNA and protein levels in periodontal membrane cells under hypoxic induction. This finding suggests that hypoxia could activate the Wnt/β-catenin signaling pathway, although the specific action mechanisms require further exploration.

The activation of the Wnt signaling pathway in periodontal stem cells is triggered by orthodontic stress through the stimulation of Tcf and the hypoxic environment, thereby modulating osteogenic differentiation. This understanding of the pivotal role of this pathway in PDLSCs offers a novel perspective for enhancing tooth movement during orthodontic interventions and maintaining the stability of alveolar bone remodeling post-treatment. We look forward to having more precise targets for the Wnt pathway in the future and to solving the difficult problems of clinical orthodontic treatment, such as long treatment time and many complications by promoting the osteogenic differentiation of PDLSCs.

Biological materials

As implant technology advances, the utilization of implants or biomaterials for mending periodontal tissue has emerged as a prominent focus in tissue engineering. A key factor for successful implantation is establishing conducive conditions to attract stem cells to the implant surface. Research indicates that leveraging PDLSCs can effectively establish a regenerative microenvironment that fosters osteogenic differentiation and augments tissue restoration around or at the implant surface, which benefits from the excellent proliferation and osteogenic differentiation potential of PDLSCs (Abo El-Dahab et al., 2024). Enhancing the osteogenic potential of PDLSCs concerning oral implants or biomaterials is pivotal for expediting bone defect repair and mitigating associated inconveniences. Consequently, a growing number of researchers are dedicated to pioneering novel materials with heightened biological efficacy, with the goal of modulating the osteogenic differentiation of PDLSCs and facilitating bone defect restoration. Given that many dental biomaterials directly interact with the periodontal ligament and alveolar bone, comprehending their impact on the osteogenic differentiation of PDLSCs serves as an essential theoretical foundation prior to the clinical implementation of these materials.

The microenvironment provided by different biomaterials will affect the osteogenic differentiation ability of PDLSCs through different signaling pathways, and surface characteristics, such as morphology, roughness, surface energy and hydrophilicity, as well as the different loads, have different effects on bone induction (Marconi et al., 2020). Boyan et al. (2018) found that the surface morphology and chemistry of materials can affect the osteogenic differentiation of mesenchymal stem cells by regulating Wnt signals. Cells on the rough surface of titanium (Ti), a commonly used implant material in clinical practice, showed increased mRNA expression of classical Wnt signaling molecules Wnt3a and β-catenin. The smooth surface affects calcium-dependent molecules Wnt5a, calmodulin and NFATc1; that is, the smooth surface of the surface can regulate the osteogenic differentiation of PDLSCs through the Wnt/ Ca2+ pathway, while the rough surface of the base can induce bone through the Wnt/β-Catenin pathway (Kim et al., 2014). A previous study on the osteoblast differentiation ability of PDLSCs found that the relative expression of Runx2 increased in the enamel matrix derivative (EMD) scaffolds compared with the blank group (Yu et al., 2020). The application of Wnt signaling pathway inhibitor DKK1 can effectively down-regulate the expression of CaMKII, NLK, Runx2 and up-regulate β-Catenin. It is speculated that the microenvironment provided by this scaffold material can affect the osteogenic differentiation potential of PDLSCs through non-classical Wnt/CaMKII/NLK signaling pathway and classical Wnt pathway. Tetrahedral DNA nanostructure (TDN) has emerged as a promising tool in tissue engineering, particularly in the realm of periodontal tissue repair. Recent research has demonstrated that TDN can significantly enhance the proliferation of PDLSCs. Furthermore, the introduction of TDN resulted in a notable 50% increase in the expression levels of osteogenic genes, such as ALP, Runx2, and OPN, just seven days post-material introduction (Zhou et al., 2019). These findings underscore the potential of TDN in promoting osteogenic differentiation and advancing periodontal tissue regeneration. Additionally, Western blot analysis has revealed heightened expression of β-catenin, suggesting that TDN holds promise as a valuable material for periodontal tissue regeneration by fostering osteogenesis through the Wnt signaling pathway. This revelation not only reinforces the potential of TDN in enhancing bone formation but also presents a novel prospect for advancing future strategies in periodontal tissue repair. In the future, more bioactive materials will promote the osteogenic differentiation of PDLSCs in the periodontal ligament by activating signaling pathways to repair tissue defects, so as to guide clinical use and improve and accelerate the clinical osseointegration process of dental implants.

Summary and outlook

PDLSCs have strong osteogenic potential and are the focus of periodontal tissue repair research. However, different environments affect the osteogenic differentiation of stem cells. There have been no specific reports in the literature on the different oral treatment situations involved in stem cell therapy. At present, many measures have been studied to promote the osteogenic differentiation of PDLSCs, such as the synthesis of polymer nanoparticles doped with doxycycline (Qiu et al., 2023) or mineralized dialgae (cu-db) (Gao et al., 2023) loaded with copper (II) ions to promote the osteogenic differentiation of hPDLSCs. There have also been studies using existing materials such as graphene quantum dots (Zong et al., 2023), gold nanocomplexes (aunc) (Gu & Bai, 2023), and natural substances such as crocin (Wu et al., 2023) and irisin (Zhang et al., 2023b) to promote bone formation. The Wnt pathway, an important signaling pathway of biological growth and development, plays an important role in the osteoblastic differentiation of PDLSCs, and its influence on osteoblastic differentiation is different in different environments. In the inflammatory microenvironment, TNF-α may negatively regulate the osteogenic capacity of PDLSCs through the classical Wnt pathway. Additionally, IL-33 and high doses of IL-1β inhibit the Wnt signaling pathway, consequently suppressing the osteogenic differentiation potential of PDLSCs. Meanwhile, IL-17 modulates the osteogenic capabilities of PDLSCs through the crosstalk between the NF-κB and Wnt signaling pathways. Furthermore, orthodontic mechanical stress and the distinct microenvironments provided by biomaterials exert varying effects on the osteogenic differentiation potential of periodontal ligament stem cells via the Wnt signaling pathway (Table 1). Targeted modulation of the Wnt signaling pathway to enhance the osteogenic differentiation of PDLSCs is poised to pave the way for the development of pharmaceuticals aimed at treating bone defects, periodontal diseases, and orthodontic complications in the realm of oral and maxillofacial surgery. Therefore, understanding how Wnt signaling pathway affects the osteogenic differentiation of PDLSCs in inflammation, orthodontics, and various biological materials has high research value and application prospect, and provides target basis for future oral clinical diagnosis and treatment. In recent years, more studies have attempted to activate the Wnt pathway through different genes to target the osteogenic differentiation of PDLSCs. Studying the effect of Wnt signaling pathway on the osteogenic differentiation of PDLSCs in different environments provides a new strategy for the repair and regeneration of orthodontic and periodontal tissues in the future. The study of Wnt signaling pathway also opens up new ideas for pharmacological regulation of PDLSC osteogenic differentiation in periodontal tissue engineering.

Table 1 Different environmental factors affect Wnt pathways.

Condition of action	Influencing factor	
TNF-α	Phosphorylated GSK-3β	
IL-17 IL-33	Activation of IKK-NF-κB to degrade β-catenin	
Orthodontic stress	Tcf	
Ti	Wnt5, NFATc1, Wnt3a, β-catenin	
EMD	CaMKII/NLK	

Although existing studies have demonstrated the indispensable role of the oral environment in investigating the osteogenic differentiation of stem cells, the findings have not yet addressed the current clinical challenges of prolonged bone remodeling times, high difficulty, and low efficiency. Therefore, there is an urgent need to apply this theoretical knowledge to develop suitable carriers or product materials that can act on the human body. This can be achieved by precisely targeting and upregulating factors that promote the osteogenic differentiation of stem cells to resolve various clinical issues related to bone repair. Future research should focus on conducting more in vivo experiments in bioengineering or materials science to fill the gaps in related studies.

Supplemental Information

Supplemental Information 1 For some patients with periodontal disease or undergoing orthodontic treatment or implant restoration, alveolar bone defects will bring significant inconvenience to clinical diagnosis and treatment.

In different environments, PDLSCs located in the periodontal ligament affect Wnt signaling pathway through various factors, as well as osteogenic differentiation. Created in BioRender.

Additional Information and Declarations

Competing Interests

Author Contributions

Data Availability

The authors declare that they have no competing interests.

Qi Su conceived and designed the experiments, performed the experiments, analyzed the data, prepared figures and/or tables, authored or reviewed drafts of the article, and approved the final draft.

Fengqiong Huang performed the experiments, authored or reviewed drafts of the article, and approved the final draft.

XiuLei Fang analyzed the data, prepared figures and/or tables, and approved the final draft.

Qiang Lin conceived and designed the experiments, authored or reviewed drafts of the article, and approved the final draft.

The following information was supplied regarding data availability:

This is a literature review.

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
