# Peer review of "The effect of the Wnt pathway on the osteogenic differentiation of periodontal ligament stem cells in different environments"

_PeerJ, doi:10.7717/peerj.18770_

## Round 0.1 · original submission · Major Revisions

Both the reviewers are minded to accept, but make some suggestions for changes. In particular, pay attention to defining the need for this review, expand your introduction and make sure you include all the different environments.

Reviewer 1 ·

Basic reporting

The article contains several typographical errors and inconsistencies in language, which can detract from the overall clarity and professionalism of the work such as
1. Abstract: Recheck “osteogenetic”. This word should be “osteogenic”
2. Line 246: Recheck "osteogenesis differentiation" It should be "osteogenic differentiation"
Therefore, please recheck and correct typographical errors to ensure the text is clear and professional.

Experimental design

No comment

Validity of the findings

It does not deeply indicate future research directions and further development. Based on your findings, discussion and conclusion from experiments, technologies, or clinical trials will help guide further exploration and application of your work.

Additional comments

No comment

·

Basic reporting

The authors described "The effect of Wnt pathway on osteogenic differentiation of periodontal ligament stem cells in different environments" that already published by previous researchers,
a. Authors need to state the differences from the other previous reviews and to justify to write this review in the introduction
b. the Introduction section is not enough. The authors must include additional background information

Experimental design

a. it is required to contain advantages or disadvantages of PDLSCs in different environments

Validity of the findings

a. in conclusion section, you should aggregate all information of PDLSCs in different environments,
b. I also suggest to add a graphical abstract to simplify the readers and or It would be better to present a figure that crystalize authors conclusion

Additional comments

a. Please choose a better quality image for figure 1
b. Additionally, some language and grammar issues are also found in the manuscript and some sentences need to be rewritten.

---

## Round 0.2 · Minor Revisions

I'm afraid that some key issues remain to be addressed. You must explain how you selected the studies for review, focus the writing as outlined and provide a direction of travel for the field.

Reviewer 1 ·

Basic reporting

1. It could highlight its broader implications, such as its potential applications in clinical practice or interdisciplinary collaborations, and how the findings connect to fields like bioengineering or material science.
2. It is unclear if the review is aimed primarily at researchers, clinicians, or both.
3. The introduction could better highlight the specific clinical challenges or knowledge gaps being addressed.

Experimental design

1. There is no explicit discussion about how studies were selected to avoid bias. For instance, were certain years, journals, or languages excluded?
2. Some citations appear incomplete or contain errors, such as "Error! Reference source not found," which undermines the review's credibility. Please complete them.

Validity of the findings

1. There is no clear suggestion of actionable steps, such as which experiments or studies should be prioritized to address the gaps.
2. The article lacks a clear goal in the introduction. Please indicate how the findings could address the identified clinical challenges or gaps in knowledge.

·

Basic reporting

standard

Experimental design

standard

Validity of the findings

standard

Additional comments

make sure that the ethic/ICUC and also
proof read some mesh space and also the references

---

## Round 0.3 · accepted · Accept

Thank you for attending to the final comments.